# L-Arginine reverses maternal and pre-pubertal codeine exposure-induced sexual dysfunction via upregulation of androgen receptor gene and NO/cGMP signaling

Roland Eghoghosoa Akhigbe[1,2], Oladele A. Afolabi[1], Ayodeji F. Ajayi [1] *

1 Anchor Reproductive Physiology and Bioinformatics Research Unit, Department of Physiology, Ladoke Akintola University of Technology, Ogbomoso, Oyo State, Nigeria, 2 Reproductive Biology and Toxicology Research Laboratory, Oasis of Grace Hospital, Osogbo, Osun State, Nigeria

* jy_ayodeji@gmail.com

## Abstract

### Background

Although codeine has been reported to enhance sexual activity by improving penile reflexes, it has been shown to impair fertility indices. Also, codeine impairs ovarian steroidogenesis and folliculogenesis. Nonetheless, whether or not codeine exerts an epigenetic effect remains unclear. On the other hand, arginine has been speculated to enhance penile reflexes by upregulating NO/cGMP Signaling.

### Aim

The study evaluated the effect of maternal codeine exposure and prepubertal codeine and arginine treatments on F1 male sexual function and fertility indices, as well as the outcome of F2 progenies. In addition, the epigenetic programming mechanism was also explored.

### Methods

Forty three-week-old female rats were randomized into two groups (n = 20 rats/group); the control that received 0.5 ml of distilled water and the codeine-treated that received 5 mg/kg of codeine via gavage for eight weeks. Afterward, the female rats were paired for mating with sexually mature male rats. Rats were maintained on their pre-pregnancy treatments throughout pregnancy and lactation. FI progenies from each cohort (control and codeine-treated cohorts) were weaned at three weeks and randomized into four groups; the control, codeine-treated, L-arginine-treated (300mg/kg), and codeine + L-arginine-treated (n = 10 rats/group). Administration commenced a week post-weaning and lasted for eight weeks via gavage.

### Key findings

Maternal codeine exposure did not alter body weight, but significantly reduced anogenital distance and anogenital index of F1 male offspring. Also, maternal codeine delayed

**Funding:** The author(s) received no specific funding for this work.

**Competing interests:** The authors have declared that no competing interests exist.

preputial membrane separation, impaired male sexual competence, and penile reflexes of F1 male offsprings. These were associated with reduced dopamine, gonadotropins, and testosterone levels as well as suppressed expression of androgen receptor mRNA. In addition, maternal codeine downregulated NO/cGMP signaling, impaired fertility indices, and reduced the litter size, weight, and survival of F2 progenies. These alterations were observed to be aggravated by prepubertal codeine exposure but improved by prepubertal arginine treatment.

## Significance

In conclusion, codeine programmed sexual dysfunction by suppressing the levels of dopamine and testosterone, as well as repressing the expression of androgen receptor mRNA. In addition, codeine-induced epigenetic reprogramming was expressed in the F2 offsprings as reduced litter size and weight, and survival rate. Notably, these observations were worsened by prepubertal codeine exposure, but dampened by prepubertal arginine treatment.

## Introduction

The endocrine disruption triggered by exposure to chemicals during development may alter the phenotype of the offspring. Increasing attention has been given to xenobiotics, including substances of abuse, acting on male reproductive function [1]. During the developmental stage, the placental is quite permeable and allows the exchange of materials across the placental barrier [2]. Also, the testes actively differentiate during this stage [2], hence exposures to endocrine-disrupting chemicals (EDCs) may lead to various phenotypic outcomes. More so, puberty is a sensitive period during which endocrine alterations may result in deleterious and possibly permanent effects on adult male reproductive function [3]. Although the molecular mechanisms of action of many EDCs are yet unclear, most have been reported to have anti-androgenic or oestrogenic effects [4, 5].

Codeine is a well-known substance of abuse globally. It is the most commonly used opioid and has been reported to be a gateway to the abuse of other substances [6, 7]. In animal models, codeine enhanced sexual performance via a testosterone-independent mechanism, although there was a potential risk of infertility evidenced by poor fertility outcomes [7]. Codeine has also been reported to impair spermatogenesis and induce low sperm quality in adult rabbits and rats [8, 9]. Studies have also revealed that codeine induces testicular damage by promoting testicular oxidative damage [10], suppressing testicular HER2, Ki67, and circulating testosterone [11], and upregulating caspase 3 signaling and p53/Bcl-2 pathway [10, 11]. In addition, codeine has been reported to impair ovarian steroidogenesis and folliculogenesis by inducing oxidative stress, inflammation, and apoptosis [12]. Despite these available shreds of evidence on the negative effects of codeine on male reproductive function, the transgenerational effect of codeine is not yet known. Also, the effect of codeine on dopamine, an essential modulator of male sexual activity, is not known.

On the other hand, L-arginine is a semi-essential proteinogenic amino acid that is synthesized endogenously; however, exogenous intake as a diet supplement contributes to the body's supply and militates against alterations in its metabolism [13, 14]. The intake of arginine has been shown to increase its concentrations in the plasma and tissues [13]. Although data reporting the impact of L-arginine on male reproductive function is limited, it has been established

as a substrate of nitric oxide (NO), thus by extension, up-regulates NO/cGMP signaling and promotes smooth muscle relaxation and vasodilatation [15] with likely positive impact on sexual function.

Notwithstanding these above-mentioned reports, studies reporting the influence of chronic codeine use on the phenotype of offspring due to possible epigenetic modifications are not available. The impact of L-arginine on codeine-induced developmental programming is also yet to be reported. Therefore, this study was designed to evaluate the epigenetic effect of maternal and pre-pubertal codeine exposure on F1 male puberty attainment, sexual function, and fertility indices, as well as the outcome of F2 progenies using a rat model. Also, the influence of L-arginine on possible codeine-induced epigenetic inheritance was probed. Furthermore, the roles of NO/cGMP signaling, androgen receptor (AR) gene expression, and circulating testosterone and dopamine as possible mechanisms of codeine activity were explored.

## Methods

Codeine was kindly donated by the National Drug Law Enforcement Agency (NDLEA), Nigeria. Arginine was commercially procured from Loba Chemie, Mumbai, India. Other reagents used were of analytical grade and obtained from Sigma Chemical Co., USA unless otherwise indicated.

### Experimental animals

Forty newly weaned 3-week-old female Wistar rats of comparable age were obtained from twenty dams (female parents) that had been monitored from the 17th day of gestation to determine the precise date of birth. The newly weaned female rats were housed in well-ventilated plastic cages (2 rats/ cage). The rats were provided normal rat chow and filtered tap water *ad libitum* and subjected to the natural photoperiod of 12 h light/dark cycle. All animals were humanely cared for under the Guide for the Care and Use of Laboratory Animals of the National Academy of Science (NAS) published by the National Institute of Health (NIH). The study was approved by the Ethics Review Committee, Ministry of Health, Oyo State, Nigeria.

### Experimental protocol

Rats were randomized into two groups; the control that received 0.5 ml of distilled water and the codeine-treated that received 5 mg/kg of codeine via gavage for eight weeks. Afterward, the female rats were paired for mating with sexually mature male Wistar rats that were maintained on rat chow only and not exposed to codeine (2:1; female: male per cage) for seven days. To control for the effect of parity on the study, all dams were virgins at the time of mating and only their first litter was included in the study. Mating was confirmed by the presence of sperm cells on the vaginal smear, and pregnancy was confirmed by body weight gain a week post-mating. Rats were maintained on their pre-pregnancy treatments throughout pregnancy and lactation.

All pups were delivered over 3 days. The pups were left undisturbed for the first week of life to prevent maternal stress. At 3 weeks old (weaning), male offsprings (F1 generation) of similar weights from each cohort (control and codeine-treated cohorts) were randomized into four groups; the control, codeine-treated, L-arginine-treated, and codeine + L-arginine-treated (n = 10 rats/group). Administration commenced a week post-weaning and lasted for eight weeks. The administration was via gavage. The dose of codeine used is the submaximal peak dose obtained from the dose-response curve of our pilot study as previously reported [11, 12]. L-arginine was administered at 300mg/kg as previously reported [14]. Animals were culled at 12 weeks (Fig 1).

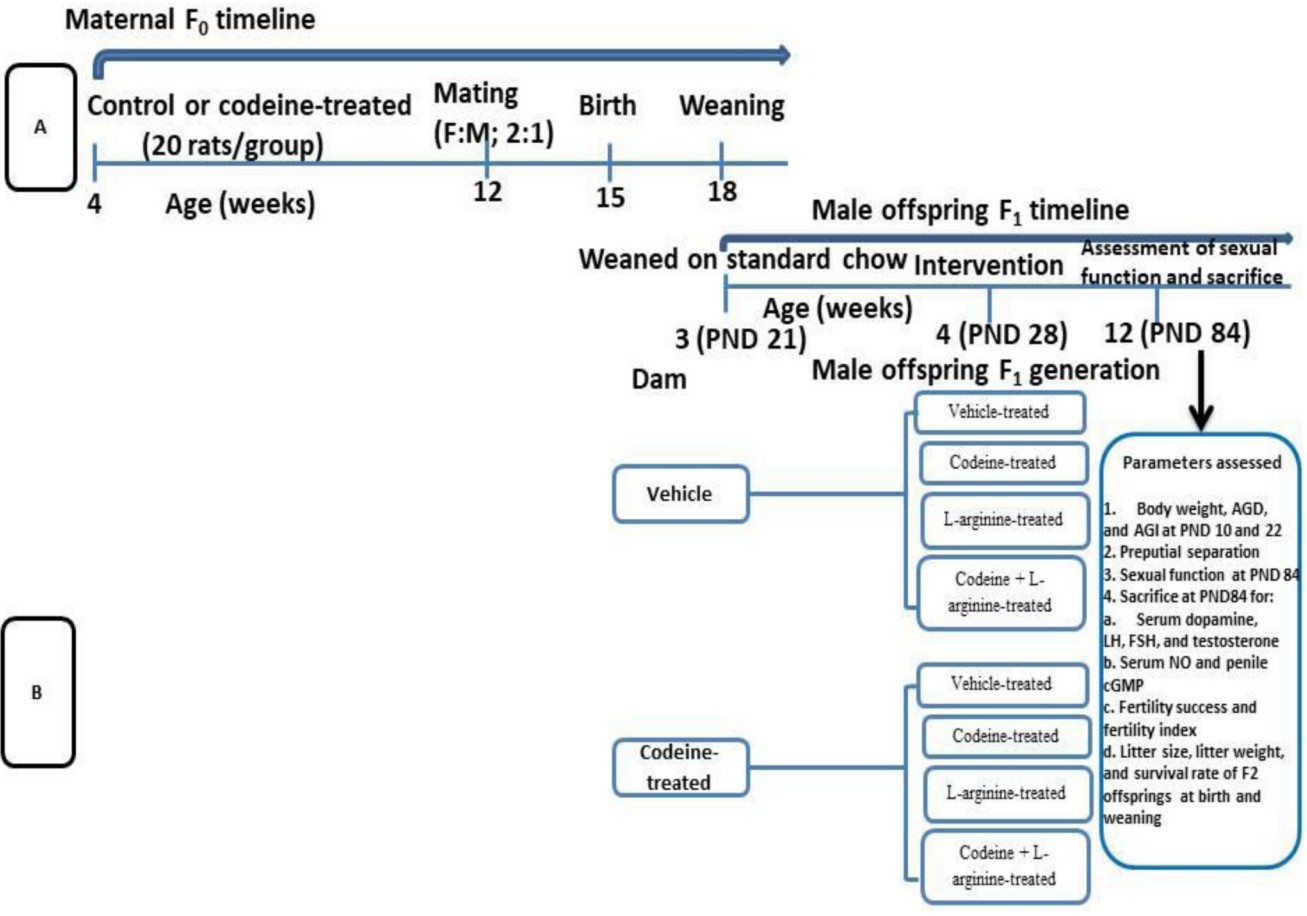

**Fig 1. Experimental treatment protocol and cohort design for $F_0$ and $F_1$ generations.**

Twenty-four hours before the sacrifice, animals were mated with age-and weight-matched female counterparts (male: female; 1: 1) that have been artificially brought to oestrous by subcutaneous administration of 10μg/100g bw of oestradiol benzoate and 0.5mg/100g bw of progesterone 48 and 4 hours respectively before mating [16]. Sexual activities were monitored for 30 minutes and recorded under dim light using a camcorder. Motivation to mate, post-ejaculation interval (PEI), and the latencies and frequencies of mount, intromissions, and ejaculations were scored by two experts, who were blinded to the study protocol as previously reported [17].

Animals were euthanized using intraperitoneal ketamine (4 mg/kg bw) and xylazine (40 mg/kg bw). Blood samples were obtained via cardiac puncture and the testis, penis, and brain were excised, trimmed off of surrounding tissues, and homogenized in cold phosphate buffer solution. Blood samples were centrifuged at 3000 rpm for 15 minutes at 4˚C to obtain the serum for biochemical assay.

## Assessment of reproductive toxicity, sexual behaviour, fertility indices, and progeny parameters

**Toxicity and endocrine disruption.** The weight of the rats was used as a marker of toxicity [18]. This was determined on postnatal day 10 (PND 10) and PND 22 (at weaning) using a sensitive electronic weighing scale. The anogenital distance (AGD) and anogenital index

(AGI) were used as a marker of endocrine disruption and reproductive toxicity [19]. These were also determined on PND 10 and PND 22. Pups were photographed digitally from the ventral aspect, including a precise 1mm scale in each picture as the reference point. The images were imported to the same computer and AGD was measured using the Image J analyzer. Measurement of each image was done twice by two different observers, who were blinded to the study protocol. The mean value of repeated measurements was used as the AGD. The AGD was measured as the distance between the base of the penis to the centre of the anal opening [2, 4], while the AGI was determined as the product of AGD and body weight$^{1/3}$ [4].

**Onset of puberty.** The preputial membrane separation was used as an index of the onset of puberty (sexual maturation) [20]. This was monitored starting from PND 24 until it was observed.

**Penile reflexes and male-type sexual behavior.** Penile reflex was assessed by the contact and non-contact methods prior to the assessment of sexual performance. The contact penile reflex was determined as the number of erections, quick flips, and long flips. Briefly, the rats were laid supine with partial restraint. The preputial sheath was pushed behind the glans and held in this position for 15 minutes to trigger genital reflexes [7]. The non-contact penile reflex was assessed as earlier reported [17]. The male rats were placed in a transparent cabin that was partitioned into two halves using sheets of plastic fibre mesh that prevented contact but permitted visual, auditory, and olfactory stimuli. After five minutes of acclimatization in the plastic cabin, each male rat was exposed to a female rat in the other half of the cabin. The visibility of the penis from the penile sheath was recorded as erection. The percentage of rats that showed penile erection by the mean number of erections was calculated as penile erection index, an index of non-contact penile reflex.

Motivation to mate, an index of libido, was scored as previously reported [17, 21]:
0: no sexual activity
1: no interaction, rears, and climbs on the chamber
2: sniffs the female rat
3: self- exploratory behaviour such as grooming and sniffing of genitals
4: grooms female counterpart anywhere
5: rears and climbs sexually
6: pursues and sniffs the female rat
7: tries to mount but is easily discouraged
8: mounts with an integrated deliberate manner and not easily discouraged
9: reflex and almost involuntary mount

Mount, intromission, and ejaculation latencies were determined as the time lags between the introduction of the female animal and the first mount by the male peer, between the introduction of the female animal and the vaginal penetration by the male counterpart, and between the first vaginal penetration and ejaculation respectively [7, 16]. Mount, intromission, ejaculation and frequencies were determined as the number of mounts from the time of introduction of the female counterpart until ejaculation, the number of vaginal penetrations from the time of introduction of the female rat to ejaculation, and the number of ejaculations from the time of the introduction of the female animal to the male within 30 min [7, 16]. PEI was determined as the interval between ejaculation and the next intromission [7].

**Fertility indices and progeny parameters.** Fertility success was determined as the percentage of the number of pregnant female rats divided by the number of paired rats, while fertility index was determined as the percentage of the number of pregnant rats divided by the number of mated rats [7].

The mean number of offsprings delivered was regarded as the litter size, while the average weight of offsprings delivered was referred to as the litter weight [17]. The survival rate at

weaning was calculated as the percentage of the number of offspring at weaning (PND 21) divided by the number of offspring at birth [17].

## Biochemical assay

**Assessment of NO/cGMP signaling.** Serum NO was assayed using Griess reaction as earlier reported [10]. Briefly, 0.1 ml of serum was added to 0.4 ml of distilled water, and then 0.1 ml of 2.5 mM sodium nitroprusside (SNP) was added. The resulting mixture was incubated at $37°C$ for 2 hours, then 0.05 ml of Griess reagent was added, and the absorbance was read at 570 nm.

The concentration of penile cGMP was determined per protein-binding principle as previously reported [22] using an ELISA kit (Elabscience Biotechnology Inc., USA) following the manufacturer's guidelines. Briefly, 50 μL of the samples and standards were pipetted into appropriate wells, while the same volume of neutralizing buffer and cGMP-HRP conjugate were pipetted into each well except for the total activity and blank wells. Also, 50 μL of 0.1M HCl was added into the non-specific binding (NSB) and the maximum binding ($B_0$) wells and 50 μL of the assay buffer were pipetted into the NSB wells. In addition, 50 μL of anti-cGMP McAb was pipetted into each well except the blank, total activity, and NSB wells. The plate was incubated at room temperature for 2 hours, and the wells were emptied and rinsed four times with the assay buffer. Five (5 μL) of the conjugate was added to the total activity wells and 150 μL of the substrate solution was added to the wells and incubated at room temperature for 10 minutes. About 50 μL of the stop solution was added to every well to halt the reaction and the plate was read immediately at 450nm absorbance.

**Male reproductive hormones.** Serum FSH, LH, and testosterone were determined using ELISA kits (Monobind Inc., USA) per the manufacturer's guidelines.

**Dopamine concentration.** The concentration of dopamine in the brain was assayed using ELISA kits (Abnova, UK) following the manufacturer's guidelines.

**AR mRNA expression.** Gene expression study was determined as earlier reported [23]. The harvested testes were homogenized in TRIzol reagent (Inqaba Biotech, South Africa). Chloroform was added to the homogenate to partition it into three phases and then centrifuged for 10 minutes. The upper layer was carefully pipetted into a clean tube. Isoamyl alcohol, the precipitating medium, was added to the solution containing the RNA pellets and vortex for 30 minutes. The supernatant was decanted to obtain the RNA pellet. Further precipitation and cleaning of RNA pellet were done by adding 70% ethanol. DNase was used to treat the sample to remove DNA contamination and obtain a DNase-free RNA. The RNA was converted to cDNA using ProtoScriptFirst Strand cDNA Synthesis Kit (NEB). AR gene was amplified by Polymerase Chain Reaction (PCR) using this primer set (5'-3'):

Forward:: `GCCATGGGTTGGCGGTCCTT`
Reverse: `AGGTGCCTCATCCTCACGCACT`
NCBI Reference Sequence: NM_012502.1

The PCR amplicon was submitted for a densitometric run in agarose 2% gel electrophoresis using TBE buffer solution (Bio Concept, Switzerland). Snapshots revealing the relative density of the DNA bands were taken under blue light documentation (Bluebox, USA). Image J software was used to quantify the intensities of the bands.

**Statistical analysis.** GraphPad Prism software was used for statistical analysis. The body weights at PND 10 and PND 22, and AGD and AGI at PND 10 and PND 22 were subjected to an unpaired student's T-test for statistical comparisons. Other data sets were compared using one-way analysis of variance (ANOVA) followed by Tukey's posthoc test for pair-wise comparison. The statistical difference was set as significant at $P < 0.05$. Data are expressed as mean ± standard deviation.

## Results

### Reproductive toxicity and endocrine disruption

Maternal codeine exposure had no significant effect on the weights of progeny at PND 10 ($p = 0.49$) and PND 22 ($p = 0.84$) (Fig 2). However, maternal codeine exposure led to a significant decrease in AGD when compared with the control on PND 10 ($p < 0.0001$) and PND 22 ($p < 0.0001$). Also, maternal codeine exposure caused a significant decrease in AGI when compared with the control on PND 10 ($p < 0.0001$) and PND 22 ($p < 0.0001$) (Fig 3).

### Preputial membrane separation

The day of preputial separation was continually monitored for the male pups from PND 24 as an index of sexual maturation and onset of male puberty. Prepubertal exposure without the maternal exposure to codeine showed no effects compared to the control. Nonetheless, maternal exposure without prepubertal exposure to codeine significantly prolonged preputial separation when compared with the control ($p = 0.001$). Also, prepubertal exposure to codeine in offsprings of codeine-exposed dams significantly worsened the effect of maternal exposure only ($p = 0.0002$). Prepubertal arginine administration ($p < 0.0001$) and arginine with codeine administration ($p < 0.0001$) significantly alleviated the effect of maternal codeine exposure on preputial membrane separation (Fig 4).

### Penile reflexes

Prepubertal codeine with or without maternal codeine exposure significantly reduced penile erection ($p < 0.0001$, $p < 0.0001$), quick flip ($p < 0.0001$, $p < 0.0001$), and non-contact penile

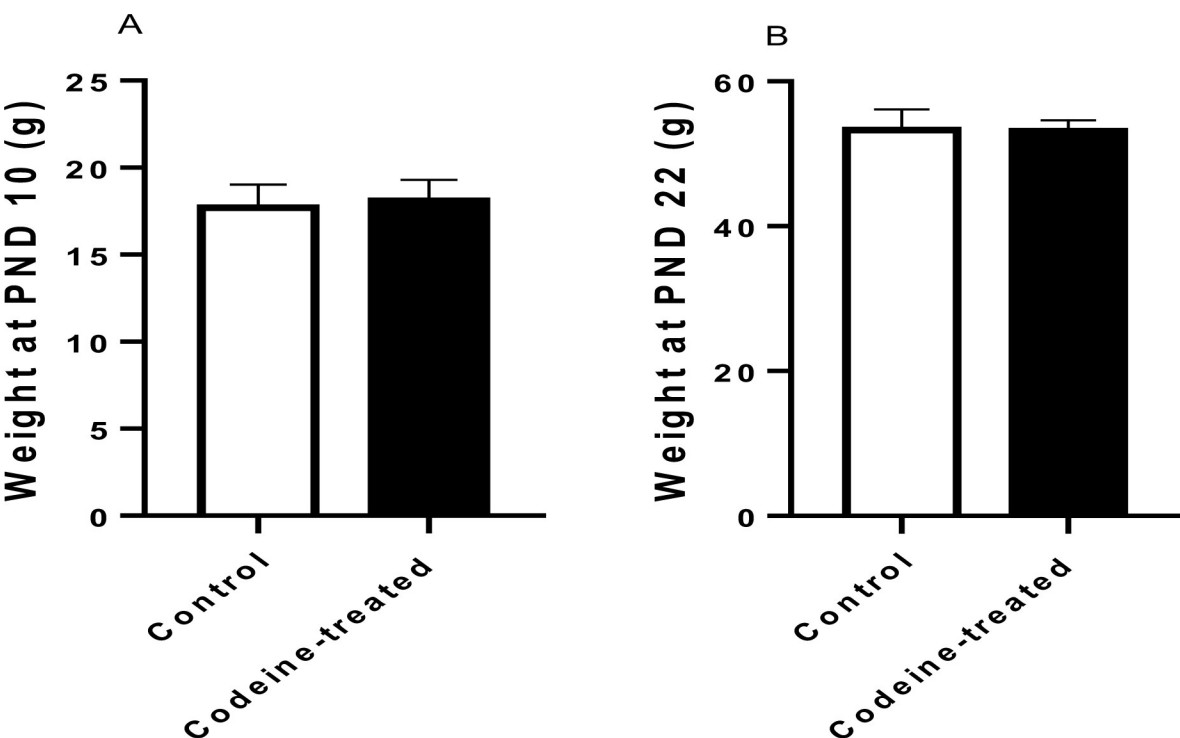

**Fig 2. Effect of codeine on body weight at post natal day (PND) 10 and 22.** Data are expressed as mean±SD for ten rats per group and analyzed by unpaired T-test.

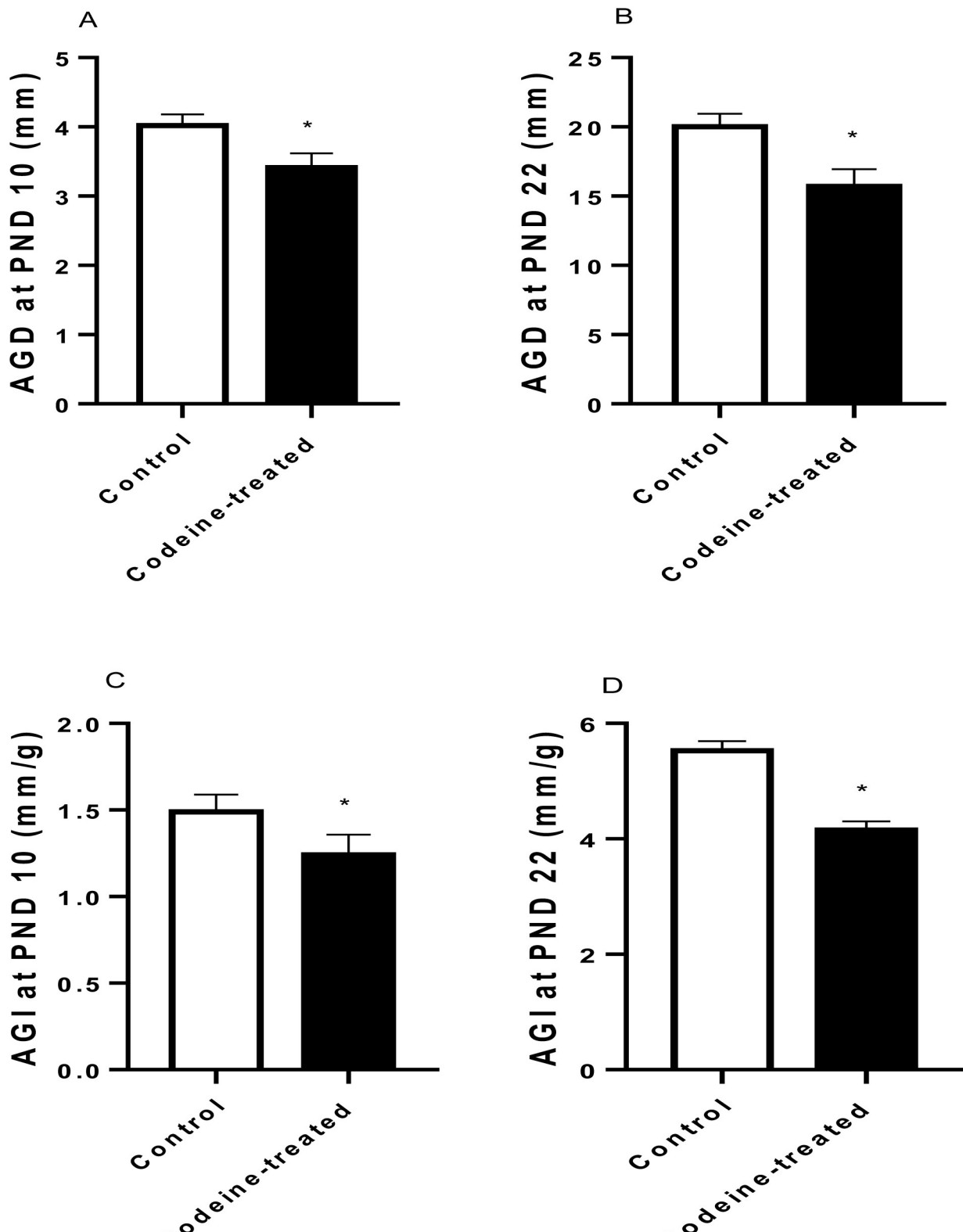

**Fig 3. Effect of codeine on anogenital distance (AGD) and anogenital index (AGI) at post natal day (PND) 10 and 22.** Data are expressed as mean±SD for ten rats per group and analyzed by unpaired T-test. *$P < 0.05$ vs control.

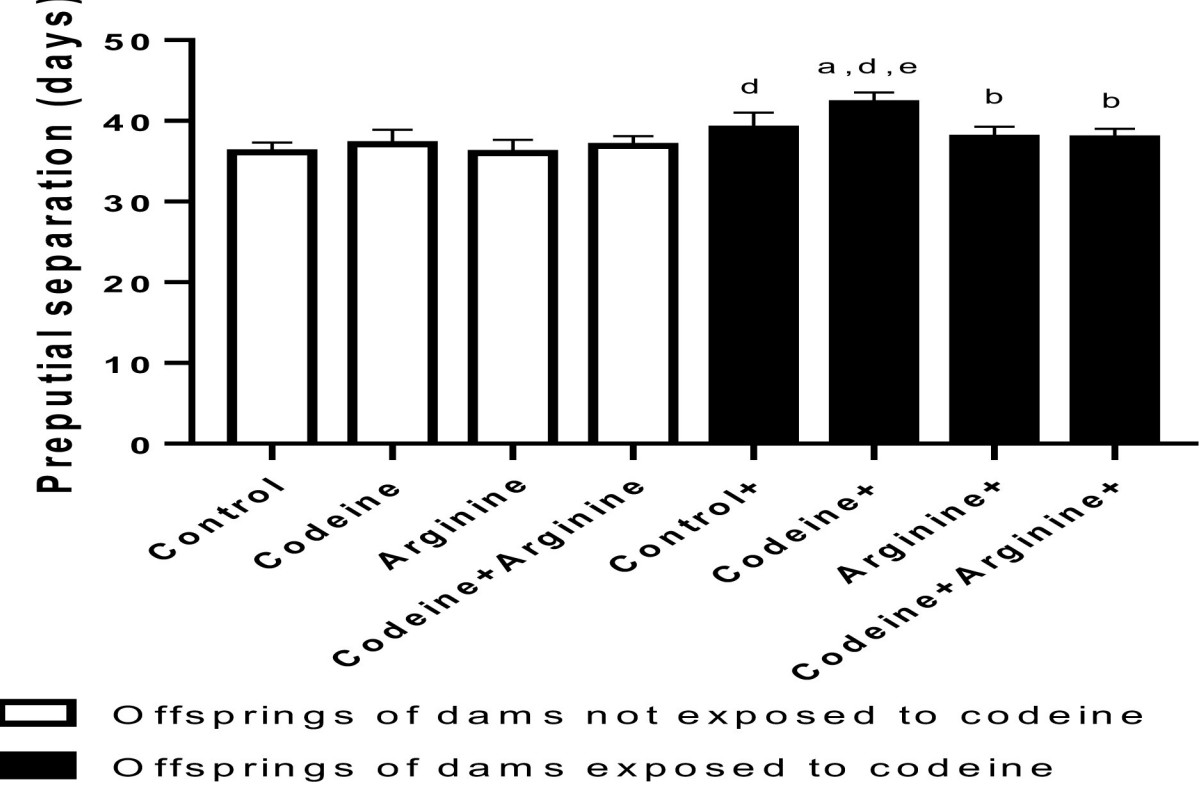

**Fig 4. Effect of maternal and prepubertal codeine use, and prepubertal arginine use on preputial membrane separation.** Data are expressed as mean±SD for ten rats per group and analyzed by one-way analysis of variance (ANOVA) followed by Tukey's posthoc test for pair-wise comparison. [a]$P < 0.05$ vs control of same cohort, [b]$P < 0.05$ vs codeine of same cohort, [d]$P < 0.05$ vs control without maternal codeine exposure, [e]$P < 0.05$ vs same prepubertal treatment without maternal codeine exposure.

reflex ($p < 0.0001$, $p < 0.0006$) when compared with the control. Although the effects of codeine on erection and non-contact penile reflex were significantly more prominent in off-springs of codeine-exposed dams compared with those whose dams were not codeine-exposed ($p = 0.020$ and $p = 0.045$ respectively), the effects of prepubertal codeine exposure on the quick flip and long flip were not altered by maternal codeine exposure ($p > 0.999$ and $p > 0.999$ respectively). More so, arginine administration significantly increased penile erection, quick flip, long flip, and non-contact penile reflex compared with the control and codeine-exposed rats, with or without maternal codeine exposure (Fig 5).

## Sexual behaviour

Overall, prepubertal codeine and arginine treatments and maternal codeine exposure influenced male sexual behaviour (Fig 6). Prepubertal codeine exposure, with or without maternal codeine exposure, led to prolonged mount, intromission, and ejaculation latencies. Also, pre-pubertal codeine exposure significantly reduced mount, intromission, and ejaculation frequencies in both cohorts (rats from dams with and without codeine exposure). These indices were not altered by maternal codeine exposure. Furthermore, arginine supplementation significantly blunted codeine-induced alterations in these male-type sexual behaviour indices.

In addition, prepubertal codeine caused a significant reduction in the motivation to mate in rats from dams without codeine exposure ($p < 0.0001$) and those from dams with codeine exposure ($p < 0.0001$) when compared with the control. This effect was not affected by

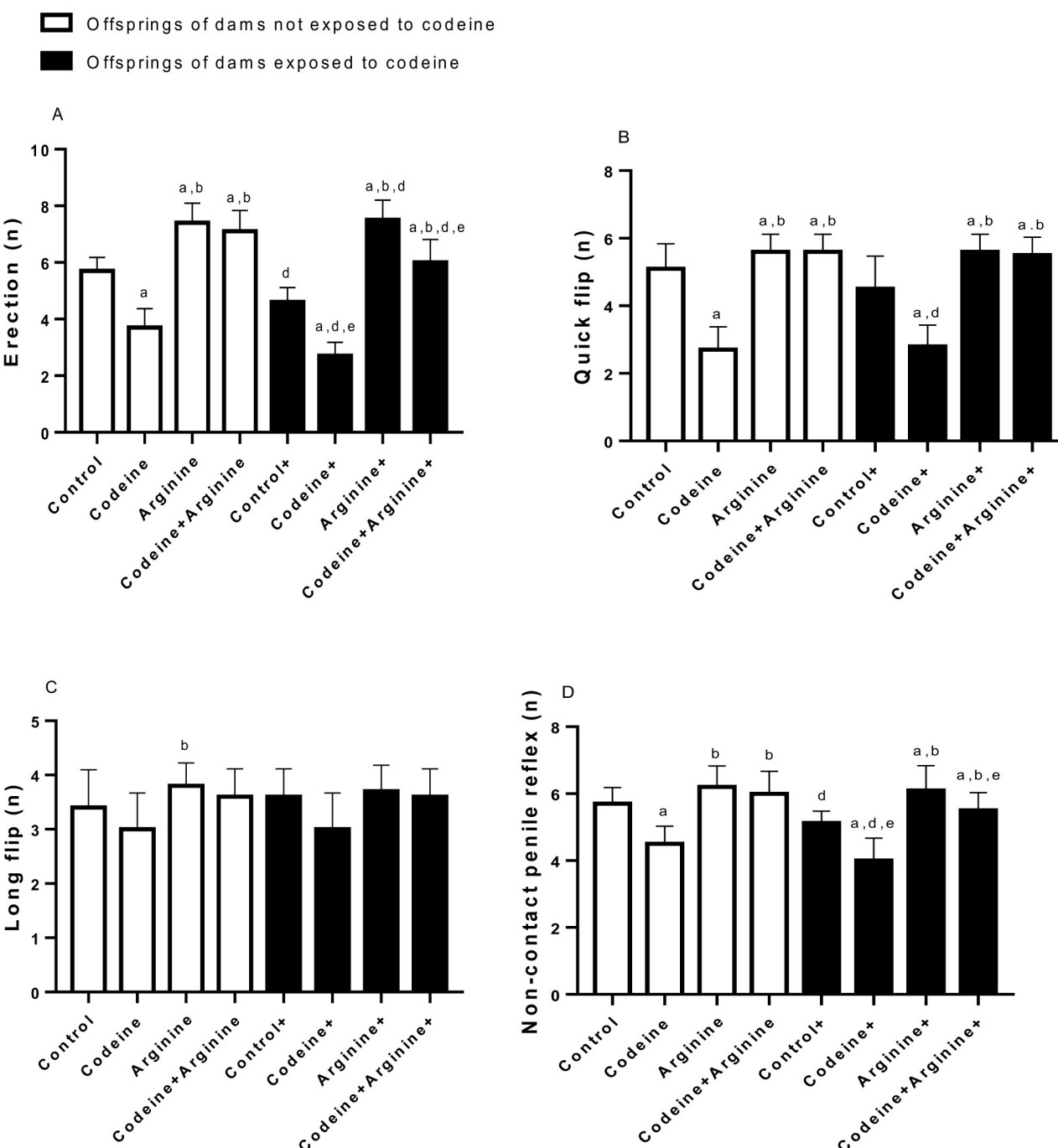

**Fig 5. Effect of maternal and prepubertal codeine use, and prepubertal arginine use on penile reflex.** Data are expressed as mean±SD for ten rats per group and analyzed by by one-way analysis of variance (ANOVA) followed by Tukey's posthoc test for pair-wise comparison. [a]$P < 0.05$ vs control of same cohort, [b]$P < 0.05$ vs codeine of same cohort, [d]$P < 0.05$ vs control without maternal codeine exposure, [e]$P < 0.05$ vs same prepubertal treatment without maternal codeine exposure.

maternal codeine exposure but was significantly abrogated by prepubertal arginine administration in rats from dams without codeine exposure ($p < 0.0001$) and those from dams with codeine exposure ($p < 0.0001$) when compared with the codeine-treated rats. Similarly, prepubertal codeine caused a significant increase in the post-ejaculation interval in rats from dams

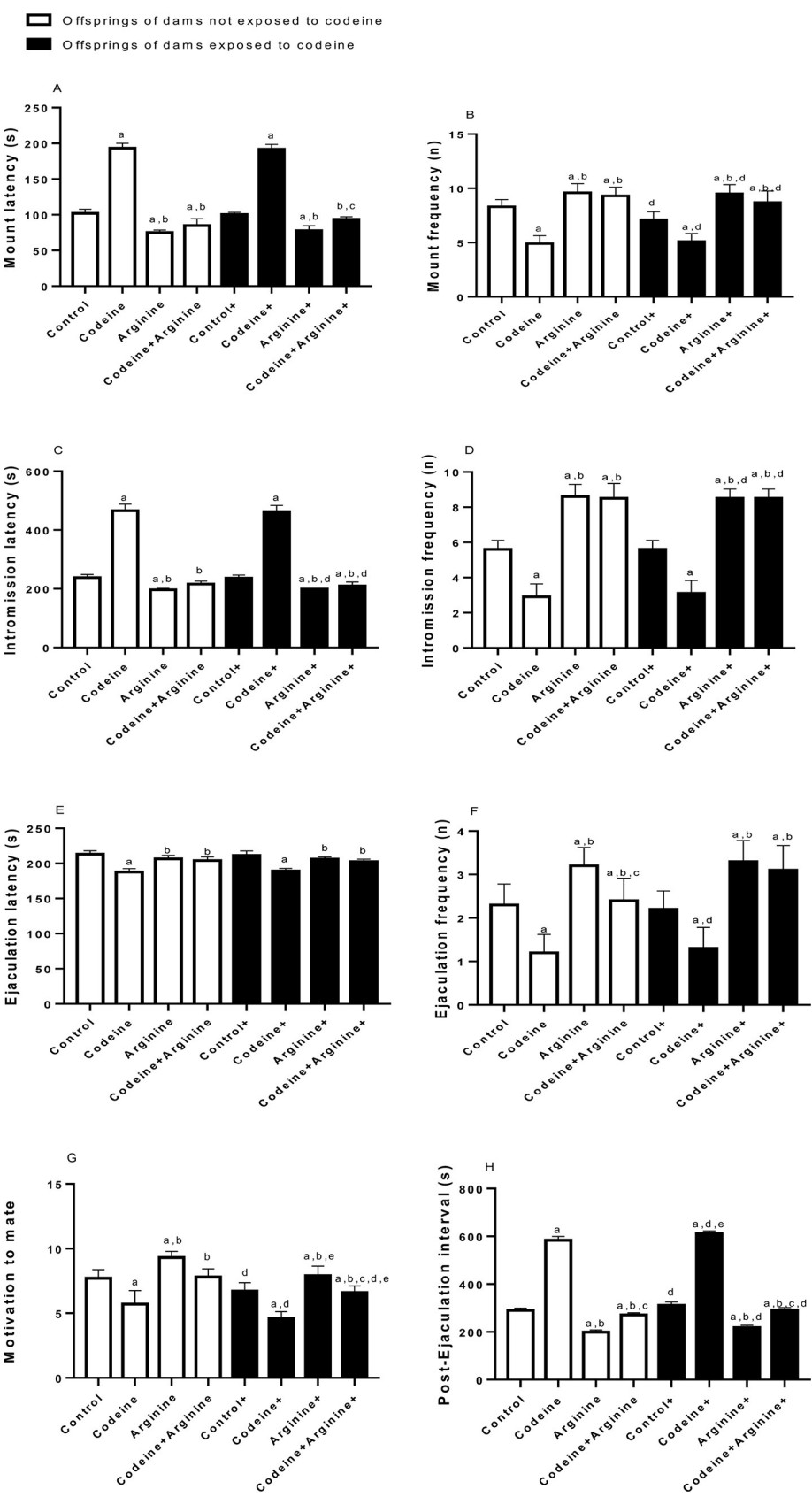

**Fig 6. Effect of maternal and prepubertal codeine use, and prepubertal arginine use on sexual urge and sexual activities.** Data are expressed as mean±SD for ten rats per group and analyzed by by one-way analysis of variance (ANOVA) followed by Tukey's posthoc test for pair-wise comparison. [a]$P < 0.05$ vs control of same cohort, [b]$P < 0.05$ vs codeine of same cohort, [c]$P < 0.05$ vs arginine of same cohort, [d]$P < 0.05$ vs control without maternal codeine exposure, [e]$P < 0.05$ vs same prepubertal treatment without maternal codeine exposure.

without codeine exposure ($p < 0.0001$) and those from dams with codeine exposure ($p < 0.0001$) when compared with the control. However, codeine-treated rats from dams that were exposed to codeine showed a significantly higher post-ejaculation interval when compared to codeine-treated rats from dams that were not exposed to codeine ($p < 0.0001$), thus suggesting that maternal exposure to codeine worsened the effect of prepubertal codeine exposure on post ejaculation interval.

## Serum NO and penile cGMP

Table 1 shows the effect of prepubertal codeine and arginine treatments and maternal codeine exposure on circulating NO and penile cGMP. Prepubertal codeine exposure led to a significant reduction in circulating NO in rats from dams without codeine exposure ($p < 0.0001$) and those from dams with codeine exposure ($p < 0.0001$) when compared with the control. Observation from this study revealed that maternal codeine exposure aggravated prepubertal codeine exposure-induced reduction in circulating NO ($p = 0.032$). Interestingly, arginine, when administered alone or with codeine, significantly improved the serum level of NO in rats from dams with and without codeine exposure. More so, prepubertal codeine exposure led to a significant reduction in penile cGMP in rats from dams without codeine exposure ($p < 0.0001$) and those from dams with codeine exposure ($p < 0.0001$) when compared with the control. The effect of prepubertal codeine exposure was noted to be significantly exacerbated by maternal codeine exposure ($p = 0.0004$). Arginine, when administered alone or with codeine, significantly improved penile cGMP levels in rats from dams with and without codeine exposure.

**Table 1. Effect of maternal and prepubertal codeine use, and prepubertal arginine use on serum nitric oxide (NO) and penile cyclic guanosine monophosphate (cGMP).**

| | Serum NO (mM) | Penile cGMP (ng/g) |
|---|---|---|
| **Control** | 0.60±0.06 | 39.6±1.50 |
| **Codeine** | 0.29±0.07 [a] | 19.50±1.58 [a] |
| **Arginine** | 0.84±0.11 [a,b] | 44.70±1.16 [a,b] |
| **Codeine + Arginine** | 0.52±0.10[b,c] | 32.50±1.43 [a,b,c] |
| **Control[+]** | 0.56±0.12 | 22.90±1.26 [d] |
| **Codeine[+]** | 0.20±0.06 [a,d,e] | 16.50±1.26 [a,d,e] |
| **Arginine[+]** | 0.55±0.08 [b,e] | 30.80±1.75 [a,b,d,e] |
| **Codeine + Arginine[+]** | 0.47±0.08[b,e] | 22.90±1.44 [a,b,c,d,e] |

[+] Offsprings of dams exposed to codeine

Data are expressed as mean±SD for ten rats per group and analyzed by unpaired T-test

[a]$P < 0.05$ vs control of same cohort

[b]$P < 0.05$ vs codeine of same cohort

[c]$P < 0.05$ vs arginine of same cohort

[d]$P < 0.05$ vs control without maternal codeine exposure

[e]$P < 0.05$ vs same prepubertal treatment without maternal codeine exposure

**Table 2. Effect of maternal and prepubertal codeine use, and prepubertal arginine use on circulating levels of gonadotropins and testosterone.**

|  | LH (mIU/ml) | FSH (mIU/ml) | Testosterone (ng/ml) |
|---|---|---|---|
| **Control** | 3.21±0.18 | 2.22±0.12 | 11.70±0.94 |
| **Codeine** | 1.57±0.11[a] | 0.48±0.07 [a] | 4.71±0.45 [a] |
| **Arginine** | 3.17±0.21 [b] | 2.25±0.13 [b] | 11.51±0.85 [b] |
| **Codeine + Arginine** | 2.85±0.15 [a,b,c] | 1.78±0.10 [a,b,c] | 9.09±0.54 [a,b,c] |
| **Control[+]** | 2.63±0.24 [d] | 1.26±0.09 [d] | 7.20±0.59 [d] |
| **Codeine[+]** | 1.27±0.12 [a,d,e] | 0.36±0.06 [a,d,e] | 4.16±0.51 [a,d,e] |
| **Arginine[+]** | 2.81±0.13 [a,b,d,e] | 1.49±0.09 [a,b,d,e] | 8.44±0.50 [a,b,d,e] |
| **Codeine + Arginine[+]** | 1.90±0.16 [a,b,c,d,e] | 0.73±0.11 [a,b,c,d,e] | 6.90±0.20 [a,b,c,d,e] |

LH: Luteinizing hormone; FSH: Follicle stimulating hormone

[+] Offsprings of dams exposed to codeine

Data are expressed as mean±SD for ten rats per group and analyzed by unpaired T-test

[a]$P < 0.05$ vs control of same cohort, [b]$P < 0.05$ vs codeine of same cohort

[c]$P < 0.05$ vs arginine of same cohort, [d]$P < 0.05$ vs control without maternal codeine exposure

[e]$P < 0.05$ vs same prepubertal treatment without maternal codeine exposure

## Male reproductive hormones

Prepubertal codeine exposure significantly reduced serum LH, FSH, and testosterone in rats from dams without codeine exposure ($p < 0.0001$) and those from dams with codeine exposure ($p < 0.0001$) when compared with the control. Arginine administration inhibited codeine-induced suppression of serum LH, FSH, and testosterone in rats from dams with and without codeine exposure (Table 2).

## Dopamine

Prepubertal codeine exposure significantly reduced the brain level of dopamine when compared with the control in rats from dams without codeine exposure ($p < 0.0006$) and those from dams with codeine exposure ($p < 0.0001$). Also, maternal codeine exposure alone significantly reduced brain dopamine levels, evidenced by a marked reduction in brain dopamine concentration in the cohort from dams with codeine exposure when compared with those from dams without codeine exposure but with the same prepubertal treatments. In addition, maternal codeine exposure worsened the effect of prepubertal codeine exposure on brain dopamine levels. However, arginine significantly improved brain dopamine levels in both cohorts (Fig 7).

## AR mRNA expression

Prepubertal exposure to codeine markedly reduced AR gene expression when compared with the control in rats from dams without codeine exposure ($p < 0.0006$) and those from dams with codeine exposure ($p < 0.0019$). Also, maternal codeine exposure significantly reduced AR gene expression, evidenced by a marked reduction in AR gene expression in the cohort from dams with codeine exposure when compared with those from dams without codeine exposure but with the same prepubertal treatments. In addition, maternal codeine exposure worsened the effect of prepubertal codeine exposure on AR gene expression. However, arginine significantly improved AR gene expression in both cohorts (Fig 8).

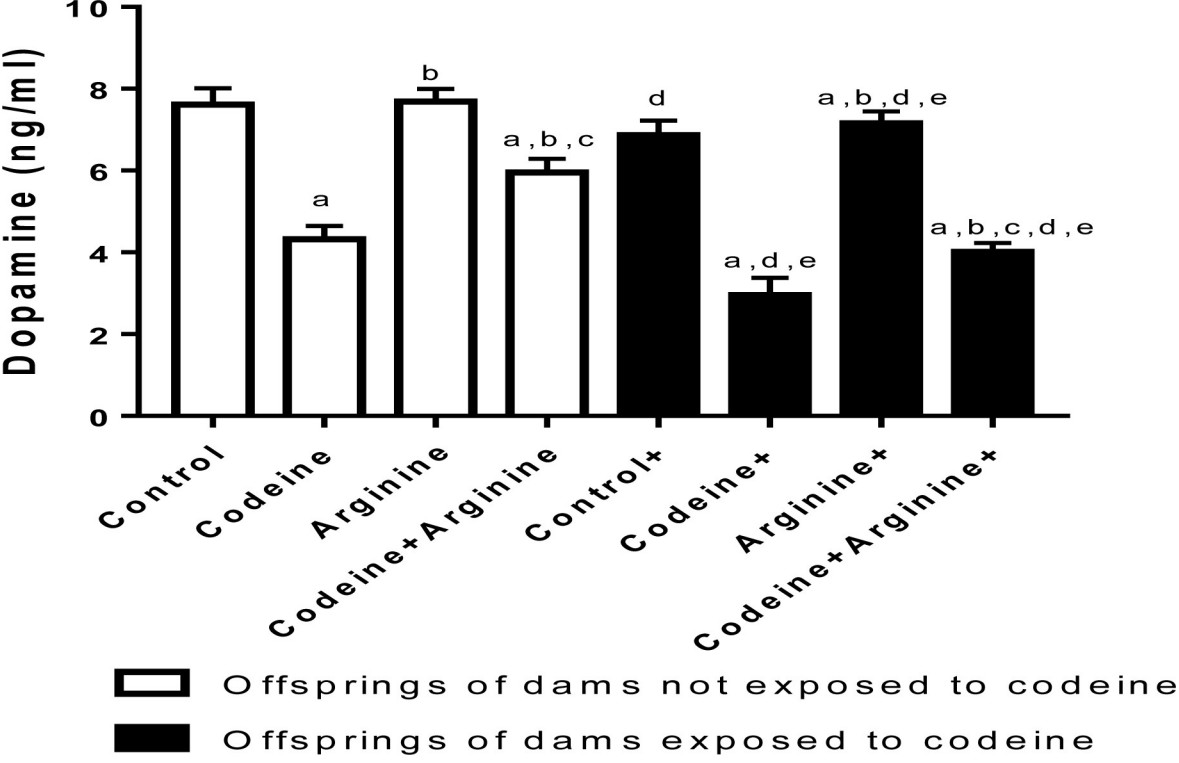

**Fig 7. Effect of maternal and prepubertal codeine use, and prepubertal arginine use on dopamine concentration.** Data are expressed as mean±SD for ten rats per group and analyzed by by one-way analysis of variance (ANOVA) followed by Tukey's posthoc test for pair-wise comparison. [a]$P < 0.05$ vs control of same cohort, [b]$P < 0.05$ vs codeine of same cohort, [c]$P < 0.05$ vs arginine of same cohort, [d]$P < 0.05$ vs control without maternal codeine exposure, [e]$P < 0.05$ vs same prepubertal treatment without maternal codeine exposure.

## Fertility indices and progeny parameters

Table 3 shows the effect of prepubertal codeine and arginine treatments and maternal codeine exposure on fertility profile. Rats with prepubertal arginine treatment with or without maternal codeine exposure had a similar fertility index (100%) but a higher fertility success (100% to 90% in control) than the control rats from dams without maternal codeine treatment. Also, when compared with the corresponding control from each cohort, prepubertal codeine exposure reduced the fertility success in rats from dams with (20%) and without (40%) codeine exposure. Similarly, when compared with the corresponding control from each cohort, prepubertal codeine exposure reduced the fertility index in rats from dams with (50%) and without (70%) codeine exposure. Maternal exposure to codeine aggravated the impact of prepubertal codeine on fertility success and fertility index. Interestingly, arginine treatment improved the fertility success and fertility index in codeine-exposed animals in both cohorts.

Although the litter size at birth was comparable across the groups except in prepubertal arginine-treated animals that showed a significant increase in litter size compared to other groups, prepubertal codeine exposure in rats birthed by codeine-exposed and codeine-unexposed dams caused a significant decrease in litter size at weaning when compared with their controls (p < 0.0001). Arginine treatment in codeine-exposed rats did not significantly alter the codeine-induced reduction in litter size at weaning (Table 4). In addition, prepubertal codeine exposure significantly reduced the litter weight at birth when compared with the control in those birthed by dams without codeine exposure (p < 0.0001) and those birthed by dams with codeine exposure (p < 0.0001). The effect of prepubertal codeine exposure on litter

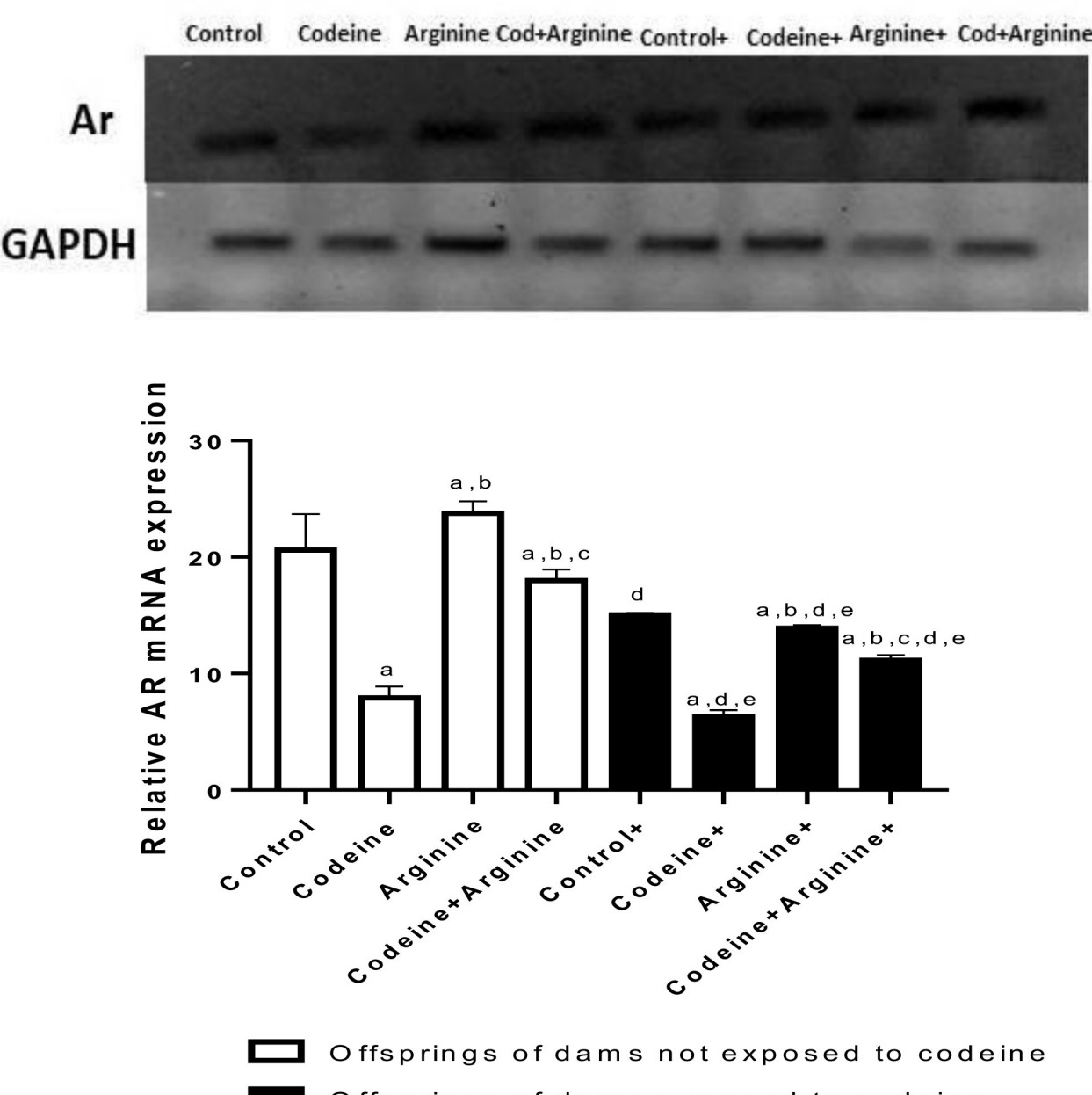

**Fig 8. Effect of maternal and prepubertal codeine use, and prepubertal arginine use on androgen receptor (AR) gene expression.** Data are expressed as mean±SD for ten rats per group and analyzed by by one-way analysis of variance (ANOVA) followed by Tukey's posthoc test for pair-wise comparison. [a]$P < 0.05$ vs control of same cohort, [b]$P < 0.05$ vs codeine of same cohort, [c]$P < 0.05$ vs arginine of same cohort, [d]$P < 0.05$ vs control without maternal codeine exposure, [e]$P < 0.05$ vs same prepubertal treatment without maternal codeine exposure.

weight at birth was ameliorated by arginine co-treatment in those birthed by dams without codeine exposure (p < 0.0001) and those birthed by dams with codeine exposure (p < 0.0001), but worsened by maternal codeine exposure (p < 0.0001). Furthermore, prepubertal codeine exposure significantly reduced the litter weight at weaning when compared with the control in those birthed by dams without codeine exposure (p < 0.0001) and those birthed by dams with codeine exposure (p < 0.0001). The effect of prepubertal codeine exposure on litter weight at weaning was blunted by arginine co-treatment in those birthed by dams without codeine

**Table 3. Effect of maternal and prepubertal codeine use, and prepubertal arginine use on fertility indices.**

|  | Fertility success (%) | Fertility index (%) |
|---|---|---|
| **Control** | 90 | 100 |
| **Codeine** | 40 | 70 |
| **Arginine** | 100 | 100 |
| **Codeine + Arginine** | 80 | 90 |
| **Control[+]** | 70 | 90 |
| **Codeine[+]** | 20 | 50 |
| **Arginine[+]** | 100 | 100 |
| **Codeine + Arginine[+]** | 90 | 100 |

[+] Offsprings of dams exposed to codeine

Data are expressed as mean±SD for ten rats per group and analyzed by unpaired T-test

[a]$P < 0.05$ vs control of same cohort, [b]$P < 0.05$ vs codeine of same cohort

[c]$P < 0.05$ vs arginine of same cohort, [d]$P < 0.05$ vs control without maternal codeine exposure

[e]$P < 0.05$ vs same prepubertal treatment without maternal codeine exposure

exposure ($p < 0.0001$) and those birthed by dams with codeine exposure ($p < 0.0001$), but worsened by maternal codeine exposure ($p < 0.0001$) (Table 4).

Addedly, prepubertal codeine exposure significantly reduced the survival rate at weaning when compared with the control in those birthed by dams without codeine exposure ($p < 0.0001$) and those birthed by dams with codeine exposure ($p < 0.0001$). The effect of prepubertal codeine exposure on survival rate at weaning was not significantly altered by arginine co-administration in those birthed by dams without codeine exposure ($p = 0.8985$) and those birthed by dams with codeine exposure ($p = 0.1338$), but worsened by maternal codeine exposure ($p = 0.0035$) (Table 4).

## Discussion

This systematic evaluation of the impacts of maternal codeine exposure on F1 generation male rats and prepubertal codeine and arginine administration on male sexual function revealed a

**Table 4. Effect of maternal and prepubertal codeine use, and prepubertal arginine use on litter size, litter weight, and survival rate.**

|  | Litter size (number) | | Litter weight (g) | | Survival rate at weaning (%) |
|---|---|---|---|---|---|
|  | At birth | At weaning | At birth | At weaning |  |
| **Control** | 8.00±0.81 | 8.00±0.81 | 6.64±0.27 | 57.32±0.90 | 96.25±6.03[a,] |
| **Codeine** | 7.00±0.81 | 5.00±0.82[a,] | 4.49±0.20[a] | 51.48±1.33[a] | 72.85±10.54 [a] |
| **Arginine** | 9.00±0.81[b] | 9.00±0.80[b] | 6.73±0.32[b] | 57.30±0.94[b] | 96.66±5.37 [b] |
| **Codeine + Arginine** | 8.00±0.81 | 6.00±0.82[a,c] | 6.05±0.33[a,b,c] | 55.00±0.82[a,b,c] | 76.25±9.22 [a,b,c] |
| **Control[+]** | 7.00±0.82 | 6.00±0.79[d] | 5.77±0.0.26[d] | 56.10±0.99 | 79.28±11.02 [a,d] |
| **Codeine[+]** | 8.00±0.81 | 4.00±0.83[a,d,e] | 3.35±0.21[a,d,e] | 48.90±1.19[d, e] | 56.25±8.83 [a,d,e] |
| **Arginine[+]** | 8.00±0.81 | 7.00±0.81[b,e] | 5.82±0.29[b,d,e] | 55.00±0.81[b,d,e] | 81.25±12.15 [a,b,e] |
| **Codeine + Arginine[+]** | 8.00±0.82 | 5.00±0.80[d] | 4.76±0.24[a,b,c,d,e] | 57.10±0.87[b,c,e] | 67.50±8.74 [a,b,e] |

[+] Offsprings of dams exposed to codeine

Data are expressed as mean±SD for ten rats per group and analyzed by unpaired T-test

[a]$P < 0.05$ vs control of same cohort, [b]$P < 0.05$ vs codeine of same cohort

[c]$P < 0.05$ vs arginine of same cohort, [d]$P < 0.05$ vs control without maternal codeine exposure

[e]$P < 0.05$ vs same prepubertal treatment without maternal codeine exposure

novel mechanism of epigenetic transgenerational inheritance. The present findings show that chronic maternal codeine exposure alters male sexual function via endocrine disruption by suppressing circulating testosterone and AR gene expression. Maternal codeine exposure adversely affected AGD and AGI, preputial membrane separation, penile reflex, sexual loco-motor activities, fertility indices, and progeny variables. These observations were aggravated by prepubertal codeine exposure and coupled with downregulation of brain dopamine levels and NO/cGMP signaling. Furthermore, this study was the first to evaluate the impact of argi-nine supplementation on the epigenetic transgenerational effects of maternal codeine exposure and prepubertal codeine exposure.

It was observed that although maternal codeine exposure did not alter body weight, it sig-nificantly reduced AGD and AGI in the F1 male generation compared with the vehicle-treated control. This result indicates that codeine is an endocrine disruptor and induces reproductive toxicity [2, 4, 19]. The observation that codeine did not alter body weight in this study agrees with our previous studies that reported the sparing effect of codeine on body weight [24–26]. Although there are no previous studies on the effect of codeine on AGD and AGI, Slamberova et al. [27] demonstrated that morphine, an opioid like codeine, reduced AGD. AGD and AGI are stable anatomical indices that reveal androgen action during the foetal testis developmental period in rodents and humans [28]. Thus, the present observation on the impact of maternal codeine exposure on AGD and AGI could imply that codeine is an endocrine-disrupting chemical, resulting in impaired foetal testicular development.

Notably, maternal codeine exposure also delayed preputial membrane separation. Preputial membrane separation is an early index of the progression of puberty [29] that precedes other markers of puberty such as motile sperm formation. It is hormone-dependent, non-invasive, and reliable in detecting chemical-induced effects on puberty attainment in rats [30]. The observed delay in preputial membrane separation in male offspring birthed by dams that were exposed to codeine implies that codeine may prolong puberty attainment and sexual matura-tion. This finding aligns with the observation of Barenys et al. [31] that revealed that 3,4-methylenedioxymethamphetamine (MDMA or "ecstasy"), an illicit drug that is also com-monly abused, delayed preputial membrane separation in rats. Also, several studies have shown that maternal under-nutrition, a form of prenatal stress, delayed preputial membrane separation [32, 33]. In addition, the current finding of the effect of codeine on preputial mem-brane separation agrees with the report of Hernández-Arteaga and his colleagues on the effect of prenatal stress on preputial membrane separation [34]. Hernández-Arteaga et al. [34] reported that preputial membrane separation was delayed following prenatal stress. Since pre-putial membrane separation is hormone-specific, this effect of codeine may be ascribed to its endocrine disrupting action on testicular development. Remarkably, prepubertal arginine treatment blunted the effect of codeine on preputial membrane separation. This may suggest that arginine may promote sexual maturation.

Several studies have documented that maternal exposure to environmental toxicants and drugs of abuse may trigger endocrine disruption and developmental toxicity with reproductive health consequences [35, 36], however, no study has reported the transgenerational impacts of maternal codeine exposure on male sexual function. The current study is the first to demon-strate that maternal codeine exposure significantly induced sexual dysfunction in an animal model evidenced by prolonged mount and intromission latencies, reduced ejaculation latency, mount, intromission, and ejaculation frequencies. This was associated with reduced motiva-tion to mate and increased post-ejaculation interval, coupled with reduced penile reflexes. These observations were worsened by prepubertal codeine exposure in the F1 male offspring. Even though there is no available data in the literature on the effect of maternal codeine expo-sure on male sexual function, the present observations on the effect of maternal codeine

exposure on male sexual indices in F1 male offspring conflict with our previous findings that documented that six weeks codeine exposure to adult male rabbits facilitates sexual locomotor activities. The disparity observed in the present study may be due to the period and duration of exposure. The exposure period was in adulthood and the duration was six weeks in the previous study, while the exposure time in the present study was in the prenatal and pubertal phases of life and the duration was eight weeks.

The frequencies of mounts and intromissions are indices of and are positively correlated to libido and sexual vigour [7, 17, 37], while latencies of mount and intromission are pointers to and negatively associated with sexual arousal [7, 38]. Also, intromission frequency is a marker of and positively associated with erection efficiency and efficacy of ejaculatory reflexes [17, 39]. In addition, the post-ejaculatory interval is a physiological marker of and negatively associated with sexual vigour [7, 17, 40]. The present findings on sexual function suggest that prenatal and prepubertal codeine exposures, unlike adulthood codeine exposure, impair sexual function and penile reflexes. Worthy of note, prepubertal arginine treatment in F1 male offspring birthed by mothers with and without codeine exposure restored sexual function and penile reflexes to near normal.

The possible roles of NO/cGMP, dopamine, testosterone, and AR gene expression in maternal codeine-induced sexual dysfunction were investigated. It was found that maternal codeine exposure led to downregulation of NO/cGMP and suppression of dopamine, FSH, LH, testosterone, and AR gene expression in the F1 male rats. These effects were worsened by prepubertal codeine exposure in the rats. Though this is the first study that evaluated the effect of codeine on NO/cGMP, dopamine, and AR gene expression, the present findings align with and corroborate our previous finding that codeine suppresses circulating testosterone levels [7, 10, 11]. It also agrees with previous reports that tramadol, an opioid that is also commonly abused, suppresses the hypothalamic-pituitary-testicular axis and reduces serum testosterone [41, 42]. It is plausible to infer that codeine-induced reduction in circulating testosterone level is attributable to its endocrine disrupting activity. Therefore, codeine reduces AGD and AGI, delays sexual maturity, and impairs reproductive function by suppressing the hypothalamic-pituitary-testicular axis, which culminates in reduced serum testosterone concentration.

More so, testosterone has been established to play a role in sexual urge, erectile function, and copulatory proficiency. Studies have demonstrated that testosterone improves libido [16, 43, 44] and copulatory proficiency [7, 45] via activation of the excitatory centres and stimulation of dopamine receptors, which is a primary factor in the stimulation of libido, sexual locomotor activity, and copulation proficiency [46]. Testosterone promotes the processing of relevant sensory stimuli, modulating the biosynthesis and release of neurotransmitters and their receptors in integrative areas, resulting in increased responsiveness of appropriate motor outputs [47]. Therefore, codeine-induced testosterone reduction may reduce libido and sexual vigour by inhibiting the excitatory centres. It is also credible to suggest that codeine impaired sexual function not just by suppressing testosterone levels but by downregulating AR gene expression, which is needed for testosterone actions. In addition, codeine-led reduction in testosterone level may, at least partly, account for the reduced level of dopamine, which is also needed to facilitate sexual urge, sexual locomotor activity, and penile reflexes [47].

Furthermore, testosterone has been shown to increase nitric oxide synthase [47, 48], thus upregulating NO concentration, NO/cGMP signaling, and by extension enhancing penile erection via relaxation of the smooth muscles of the cavernous arteries and tissue [49]. Interestingly, NO has been shown to increase basal and female-stimulated dopamine release, resulting in enhanced copulation and male genital reflexes [47]. Hence, codeine-induced sexual dysfunction may involve several pathways viz suppression of testosterone and AR, reduction of dopamine level, and downregulation of NO/cGMP signaling.

Noteworthy, prepubertal arginine administration depressed codeine-induced sexual dysfunction and enhanced libido, sexual drive, and penile reflexes. It is likely that arginine enhanced NOs activity and improved NO production [45]. The improved circulating NO level exerts immune-modulatory effects and inhibits the migration of monocyte and lymphocytes to the endothelium, thus maintaining penile endothelial homeostasis [45] and promoting penile reflexes. Also, arginine-driven NO increase possibly enhanced tissue perfusion, promoting their secretory function [13], and resulting in improved testosterone levels. It is also not unlikely that arginine-led NO rise activated soluble guanylyl cyclase that converts guanosine triphosphate to cGMP, which in turn activates protein kinase G to promote vasorelaxation and penile smooth muscle relaxation [15]. In addition, NO upregulates dopamine release, resulting in enhanced sexual activity and genital reflexes [47].

Besides the observation that prenatal and prepubertal codeine exposure led to sexual dysfunction, it was also noted that maternal exposure to codeine led to reduced fertility indices in F1 generation male offsprings and poor F2 progenies evidenced by reduced F2 generation litter size and weight, and decreased survival rate at weaning. These findings underscore the multigenerational epigenetic effect of codeine, and also agree with previous studies that reported that endocrine disruptors elicit developmental toxicity [35, 36]. It is likely that prenatal and prepubertal codeine exposure impairs development and induces toxicity, thus exerting a negative effect on the survival of the progenies. Astonishingly, arginine also repressed codeine-induced developmental programming in F1 male offsprings and F2 offsprings, evidenced by improved fertility indices, increased litter size and weight, and survival rate at weaning. Summing up, maternal codeine exposure in the F0 generation induced transgenerational epigenetic reprogramming of the oocyte that likely persisted during pregnancy and influenced the extracellular milieu *in utero*, which affected the F1 male offsprings as well as the primordial germ cells (that are representatives of the F2 progenies).

In conclusion, this study revealed that maternal codeine exposure exerts endocrine disruption, delayed sexual maturation, and induced sexual dysfunction in F1 male offspring. This was coupled with reduced fertility indices in the F1 male offsprings and poor foetal outcomes in the F2 progenies and worsened by prepubertal codeine exposure in the f1 male offsprings. The programming mechanism could be associated with the low-level programming of testosterone/AR genes. However, prepubertal arginine treatments repressed codeine-induced endocrine disruption, delayed sexual maturation, and sexual dysfunction. This study provides novel experimental evidence of the transgenerational epigenetic effect of codeine and the therapeutic value of arginine as a candidate molecule for ameliorating epigenetic alterations induced by codeine, and possibly other endocrine disruptors (Fig 9). Thus, studies exploring the likely epigenetic effects of codeine on other systems, including other aspects of reproductive health are recommended. Also, more experimental and clinical studies will be important to validate the therapeutic use of arginine in preventing the epigenetic impact of codeine and other endocrine disruptors.

Although this study presents compelling novel shreds of evidence on the transgenerational effects of codeine and arginine on sexual function, there were some limitations. First, no data on sperm quality and testicular histopathological changes were presented. This is because the ejaculatory activity during the assessment of sexual function could impact directly on sperm reserve, thus altering sperm quality. In addition, epigenetic variables such as markers of DNA methylation and histone modification were not included. Hence, future studies assessing sperm quality, testicular histopathological changes, and epigenetic variables are recommended.

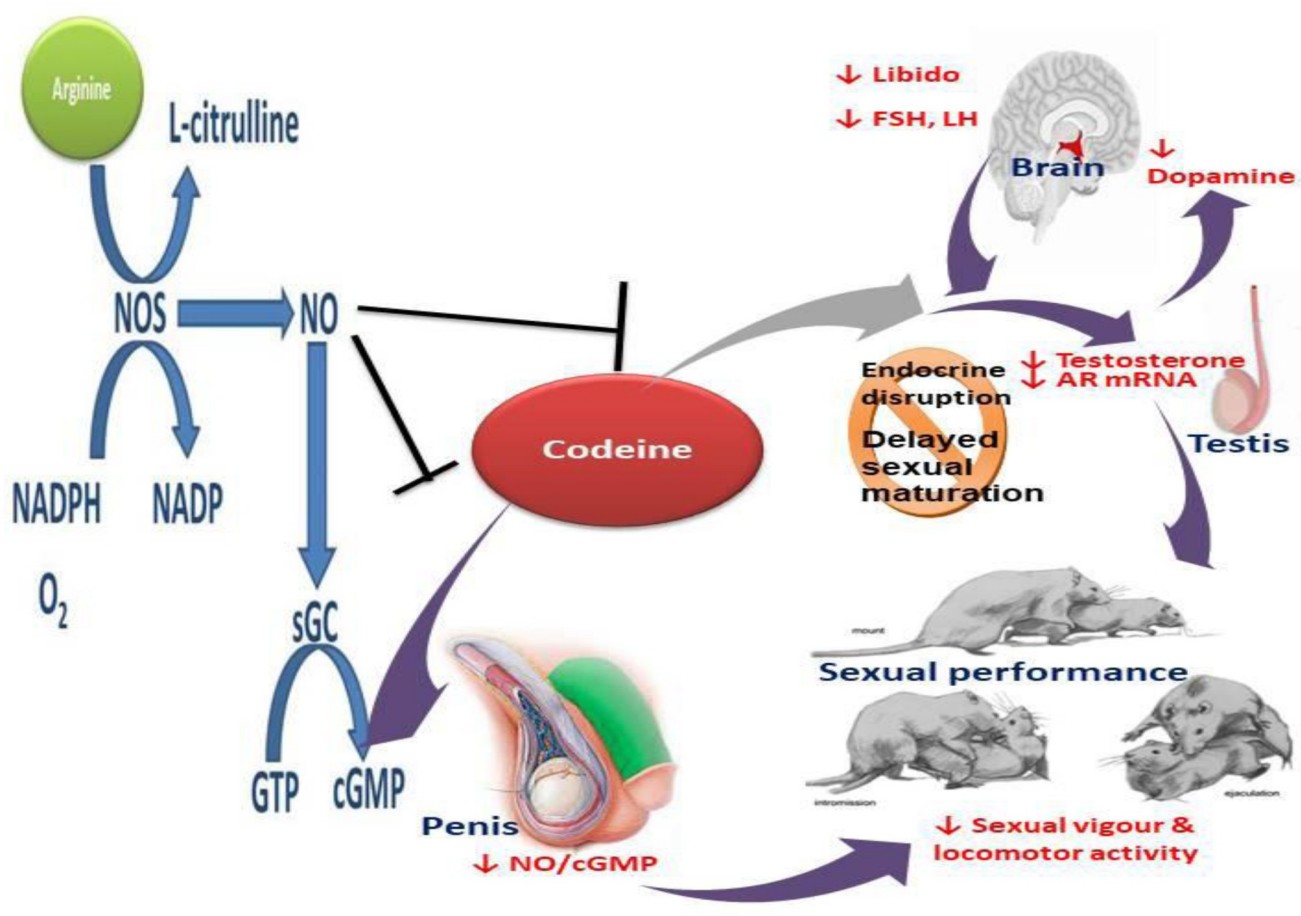

**Fig 9. Schematic representation of the effect of maternal codeine exposure and prepubertal codeine and arginine use on sexual maturation and sexual performance.**

## Supporting information

**S1 Raw images.**
(DOC)

## Author Contributions

**Conceptualization:** Roland Eghoghosoa Akhigbe, Oladele A. Afolabi, Ayodeji F. Ajayi.

**Data curation:** Roland Eghoghosoa Akhigbe, Ayodeji F. Ajayi.

**Formal analysis:** Roland Eghoghosoa Akhigbe.

**Funding acquisition:** Roland Eghoghosoa Akhigbe.

**Investigation:** Roland Eghoghosoa Akhigbe.

**Methodology:** Roland Eghoghosoa Akhigbe, Oladele A. Afolabi, Ayodeji F. Ajayi.

**Project administration:** Roland Eghoghosoa Akhigbe.

**Resources:** Roland Eghoghosoa Akhigbe, Oladele A. Afolabi, Ayodeji F. Ajayi.

**Software:** Roland Eghoghosoa Akhigbe, Oladele A. Afolabi, Ayodeji F. Ajayi.

**Supervision:** Oladele A. Afolabi, Ayodeji F. Ajayi.

**Validation:** Oladele A. Afolabi, Ayodeji F. Ajayi.

**Writing – original draft:** Roland Eghoghosoa Akhigbe.

**Writing – review & editing:** Roland Eghoghosoa Akhigbe, Oladele A. Afolabi, Ayodeji F. Ajayi.

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
