## [Decision Letter · Decision Letter 0]

8 Jun 2022

PONE-D-22-06170L-Arginine Reverses Maternal and Pre-pubertal Codeine Exposure-Induced Sexual Dysfunction via Upregulation of Androgen Receptor Gene and NO/cGMP SignalingPLOS ONE

Dear Dr. Ajayi,

Thank you for submitting your manuscript to PLOS ONE. After careful consideration, we feel that it has merit but does not fully meet PLOS ONE’s publication criteria as it currently stands. Therefore, we invite you to submit a revised version of the manuscript that addresses the points raised during the review process.

We look forward to receiving your revised manuscript.

Kind regards,

Christopher Torrens

Academic Editor

PLOS ONE

Journal Requirements:

In your cover letter, please note whether your blot/gel image data are in Supporting Information or posted at a public data repository, provide the repository URL if relevant, and provide specific details as to which raw blot/gel images, if any, are not available. Email us at plosone@plos.org if you have any questions

3. Please include your tables as part of your main manuscript and remove the individual files. Please note that supplementary tables (should remain/ be uploaded) as separate "supporting information" files.

Reviewers' comments:

Reviewer's Responses to Questions

**Comments to the Author**

1. Is the manuscript technically sound, and do the data support the conclusions?

Reviewer #1: Partly

Reviewer #2: Partly

2. Has the statistical analysis been performed appropriately and rigorously? 

Reviewer #1: Yes

Reviewer #2: N/A

3. Have the authors made all data underlying the findings in their manuscript fully available?

Reviewer #1: Yes

Reviewer #2: Yes

4. Is the manuscript presented in an intelligible fashion and written in standard English?

Reviewer #1: Yes

Reviewer #2: Yes

5. Review Comments to the Author

Reviewer #1: General comments: The manuscript brings an interesting and current subject. The experimental design, although a little confusing, makes it clear that the focus is on male reproduction and the possible programmed effects in the next generations. However, as a reproductive toxicology study, many analyzes should have been performed and were not, such as analyzes of sperm quality and fertility parameters. But the most important are the histopathological and consequently immunohistochemical analyzes for some fundamental receptors, at least for the gonads. These analyzes are of great importance due to the objectives of the study itself and of course, due to the data obtained, they could detail how some changes occurred or what were the impacts of the promoted changes, such as the reduction of hormone levels.

> The authors described all the impacts of codeine on male reproduction and sperm quality and only cite a few impacts on female reproduction. Subsequently, the experimental design shows that females were treated for 8 weeks from the beginning. These females, after get pregnancy, were no longer treated. Male pups that were subdivided and treated from the 4th postnatal week onwards. I would like to understand why females are not treated during pregnancy (best window for fetal reprogramming) and why not more emphasis on the impacts of codeine and L-arginine on females reproductive systhem.

> The males were euthanized 24 hours after performing sexual behavior. This protocol directly impacted the sperm reserves of these animals and even if they all performed sexual behavior, any change found in the ejaculatory activity could impact directly on the sperm reserves of these animals, which would not mean a reduction in production or acceleration of sperm transit time, but an increase in ejaculatory activity for example. Why wasn't 15 days waiting (ideal period) for the recovery of the sperm reserves of these animals and sperm quality analyzes performed, in addition to the proposed biochemical analyzes?

> The fertility test after sexual behavior was evaluated after birth. Correct? Why did the authors not perform the euthanasia of the pregnant females to obtain all the data, such as pre and post-implantation losses and also possible malformations, since it is known that females when they give birth, discard by eating malformed or stillborn pups, losing these data.

> Why were histological analyzes of the testes and epididymis not performed, at least, since the authors discussed about fetal programming and for toxicological assays, histophatology is one of the most important assays?

>I believe that a schematic design of the entire experimental protocol adopted would be interesting to clarify the periods and what was analyzed in each generation. I suggest that the authors add a figure 01 to the material and methods, detailing the analyzes and periods performed for the entire experimental protocol (all generations used).

> Figure 05 - this figure would be much more interesting in the form of a table, with the absolute values of each parameter.

> The authors discuss epigenetics effects, however, specific parameters were not evaluated in relation to this result, both in F1 and F2 generations.

> Figure 07 - The reduction in mRNA expression for AR is another fact that reinforces the need for analysis of sperm quality and histopathology of the gonads in this present study.

Reviewer #2: Comments：

PONE-D-22-06170: L-Arginine Reverses Maternal and Pre-pubertal Codeine Exposure-Induced Sexual Dysfunction via Upregulation of Androgen Receptor Gene and NO/cGMP Signaling.

The study evaluated the effect of maternal codeine exposure and prepubertal codeine and arginine treatments on F1 male sexual function and fertility indices, as well as the outcome of F2 progenies. In addition, the epigenetic programming mechanism was also explored, which is related to decreased dopamine analysis, decreased testosterone level, and inhibition of local GR expression in testis.In addition, when the male offspring of F1 generation were mated to F2 generation, pathological changes such as shortened fetal body length, weight loss and reduced number of live fetuses were observed.And these changes were reversed when given arginine.This study is helpful to guide the healthy life of male and to guide eugenics.In my opinion, the manuscript is innovative to a certain extent and needs to be revised accordingly. Specific opinions are as follows:

1、 It is important that the authors write this discussion without discussing the reality of codeine exposure during pregnancy or adolescence, the basis for modeling and the corresponding timing and duration of administration are also very important.

2、 When explaining the mechanism of pathological changes, the author uses AR as the research target, so what is the specific signal pathway that AR participates in this project? The author did not carry out corresponding detection, should not fill?

3、 AR is expressed in a variety of testicular cells, such as Sertoli cells, Leydig cells and epithelial cells. What kind of cells does codeine mainly affect AR? Are you considering further validation in cells?

4、 The result part of this paper is too monotonous, so the result part of the manuscript should be enriched, such as adding some morphological changes of testis, sperm quality and other related indicators;

5、 Why does this paper focus on dopamine? Is it due to the biological effects of codeine? The author needs to describe the corresponding reasons for selection in the discussion.

6、There are many irregularities in writing in the manuscript:

（1） The notation is not standardized, and no specific explanation is given for each small figure

（2） For the reference section, a unified format is required

（3） In the WB results, the sensation is not produced on the same experiment. It should be replaced.

7、There are many grammatical errors in the full text, which make it difficult to read the manuscript. Please check it.

6. PLOS authors have the option to publish the peer review history of their article (what does this mean?). If published, this will include your full peer review and any attached files.

Reviewer #1: No

Reviewer #2: No

---

## [Author Response · Author response to Decision Letter 0]

16 Jun 2022

Reviewers' comments:

Reviewer's Responses to Questions

Comments to the Author

1. Is the manuscript technically sound, and do the data support the conclusions?

Reviewer #1: Partly

Reviewer #2: Partly

Response: Thanks.

2. Has the statistical analysis been performed appropriately and rigorously?

Reviewer #1: Yes

Reviewer #2: N/A

Response: Thanks.

3. Have the authors made all data underlying the findings in their manuscript fully available?

Reviewer #1: Yes

Reviewer #2: Yes

Response: Thanks.

4. Is the manuscript presented in an intelligible fashion and written in standard English?

Reviewer #1: Yes

Reviewer #2: Yes

Response: Thanks.

5. Review Comments to the Author

Reviewer #1: General comments: The manuscript brings an interesting and current subject. The experimental design, although a little confusing, makes it clear that the focus is on male reproduction and the possible programmed effects in the next generations. However, as a reproductive toxicology study, many analyzes should have been performed and were not, such as analyzes of sperm quality and fertility parameters. But the most important are the histopathological and consequently immunohistochemical analyzes for some fundamental receptors, at least for the gonads. These analyzes are of great importance due to the objectives of the study itself and of course, due to the data obtained, they could detail how some changes occurred or what were the impacts of the promoted changes, such as the reduction of hormone levels.

Response: Thanks. The focus of the present study is on puberty attainment and sexual function, hence parameters on sperm quality and testicular histopathological changes were not included. Also, following the assessment of sexual function, sperm parameters such as sperm motility and sperm count would be affected; hence analysis of sperm parameters was not included.

In addition, gene expression was used for receptor expression (AR mRNA). 

However, fertility parameters such as fertility index and fertility success were included (Table 3).

> The authors described all the impacts of codeine on male reproduction and sperm quality and only cite a few impacts on female reproduction. Subsequently, the experimental design shows that females were treated for 8 weeks from the beginning. These females, after get pregnancy, were no longer treated. Male pups that were subdivided and treated from the 4th postnatal week onwards. I would like to understand why females are not treated during pregnancy (best window for fetal reprogramming) and why not more emphasis on the impacts of codeine and L-arginine on females reproductive systhem.

Response: Thanks. The present study was aimed at assessing the impact of maternal codeine exposure on male offsprings, hence parameters on dams or female offsprings were not included.

Also, The dams (mothers) were treated with codeine before pregnancy, at pregnancy, and during lactation. This has been justified in the first paragraph of the INTRODUCTION.

> The males were euthanized 24 hours after performing sexual behavior. This protocol directly impacted the sperm reserves of these animals and even if they all performed sexual behavior, any change found in the ejaculatory activity could impact directly on the sperm reserves of these animals, which would not mean a reduction in production or acceleration of sperm transit time, but an increase in ejaculatory activity for example. Why wasn't 15 days waiting (ideal period) for the recovery of the sperm reserves of these animals and sperm quality analyzes performed, in addition to the proposed biochemical analyzes?

Response: Thanks. The authors completely agree with this point. This explains why we did not include data on sperm parameters. This would be included in a future study (where assessment of sexual performance would not be conducted). 

> The fertility test after sexual behavior was evaluated after birth. Correct? Why did the authors not perform the euthanasia of the pregnant females to obtain all the data, such as pre and post-implantation losses and also possible malformations, since it is known that females when they give birth, discard by eating malformed or stillborn pups, losing these data.

Response: Thanks. The authors completely agree with this point. Fertility assessment includes fertility index, fertility success, litter size and weight, and survival rates at weaning. Although, pre and post-implantation losses and also possible malformations are important, they are more of teratogenic/toxicity assessment. Since the assessment of pre and post-implantation losses would require the sacrifice of the female rats (dams) before delivery, evaluation of core fertility parameters like fertility index, fertility success, litter size and weight, and survival rates at weaning would be impossible after sacrifice. Hence, we employed fertility index, fertility success, litter size and weight, and survival rates at weaning. 

> Why were histological analyzes of the testes and epididymis not performed, at least, since the authors discussed about fetal programming and for toxicological assays, histophatology is one of the most important assays?

Response: Thanks. The sexual performance would alter the sperm reserve in the epididymides and testis, hence histopathological analyses of these organs were not conducted in the present study. The current study focused on male-type sexual function programming. This would be included in a future study on sperm and spermatogenesis programming (where assessment of sexual performance would not be conducted).

>I believe that a schematic design of the entire experimental protocol adopted would be interesting to clarify the periods and what was analyzed in each generation. I suggest that the authors add a figure 01 to the material and methods, detailing the analyzes and periods performed for the entire experimental protocol (all generations used).

Response: Thanks. A figure on experimental protocol has been added as Figure 1.

> Figure 05 - this figure would be much more interesting in the form of a table, with the absolute values of each parameter.

Response: Thanks. There are 8 parameters and 8 groups in Figure 5. Presenting these in a table will look too cumbersome; such are better presented in a Figure.

> The authors discuss epigenetics effects, however, specific parameters were not evaluated in relation to this result, both in F1 and F2 generations.

Response: Thanks. Epigenetics involves the transgenerational modification without alteration of the DNA. Thus, data on male F1 offsprings explains the epigenetic effects of codeine and arginine. Markers of epigenetics were not included in the presented study but this has been included as a limitation of the study for future exploration.

> Figure 07 - The reduction in mRNA expression for AR is another fact that reinforces the need for analysis of sperm quality and histopathology of the gonads in this present study.

Response: Thanks. AR is a receptor for androgen and the provision of data on hormone corroborates the findings of reduction in mRNA expression for AR. Sperm quality and gonad histology were not evaluated in the present study because assessment of sexual function would affect sperm reserve as well as sperm quality. Future studies reporting these would be needed and has been incorporated in the recommendation. 

Reviewer #2: Comments：

PONE-D-22-06170: L-Arginine Reverses Maternal and Pre-pubertal Codeine Exposure-Induced Sexual Dysfunction via Upregulation of Androgen Receptor Gene and NO/cGMP Signaling.

The study evaluated the effect of maternal codeine exposure and prepubertal codeine and arginine treatments on F1 male sexual function and fertility indices, as well as the outcome of F2 progenies. In addition, the epigenetic programming mechanism was also explored, which is related to decreased dopamine analysis, decreased testosterone level, and inhibition of local GR expression in testis.In addition, when the male offspring of F1 generation were mated to F2 generation, pathological changes such as shortened fetal body length, weight loss and reduced number of live fetuses were observed.And these changes were reversed when given arginine.This study is helpful to guide the healthy life of male and to guide eugenics.In my opinion, the manuscript is innovative to a certain extent and needs to be revised accordingly. Specific opinions are as follows:

Response: Thanks.

1、 It is important that the authors write this discussion without discussing the reality of codeine exposure during pregnancy or adolescence, the basis for modeling and the corresponding timing and duration of administration are also very important.

Response: Thanks. This has been included. A figure showing the experimental protocol has also been added as Figure 1.

2、 When explaining the mechanism of pathological changes, the author uses AR as the research target, so what is the specific signal pathway that AR participates in this project? The author did not carry out corresponding detection, should not fill?

Response: Thanks. The authors implicated AR as a target, resulting in reduced androgen response (Page 21, paragraph 2). NO/cGMP signaling was also implicated (Page 22, paragraphs 1 and 2).

3、 AR is expressed in a variety of testicular cells, such as Sertoli cells, Leydig cells and epithelial cells. What kind of cells does codeine mainly affect AR? Are you considering further validation in cells?

Response: Thanks. Authors are considering further validation of the cells.

4、 The result part of this paper is too monotonous, so the result part of the manuscript should be enriched, such as adding some morphological changes of testis, sperm quality and other related indicators;

Response: Thanks. The focus of the present study is on puberty attainment and sexual function, hence parameters on sperm quality and testicular histopathological changes were not included. Also, following ejaculatory activities during assessment of sexual function, sperm parameters such as sperm motility and sperm count would be affected; hence analysis of sperm parameters was not included. This has been included in the study limitation (Page 24, paragraph 1).

However, indicators of the study focus were included such as:

1. Puberty attainment and endocrine disruption: Preputial separation, AGD, AGI

2. Sexual function: Mount, intromission, and ejaculation latencies and frequencies, dopamine

3. Mechanisms: Endocrine disruption (AGD, AGI, LH, FSH and testosterone as well as AR), erectile function (NO, cGMP)

5、 Why does this paper focus on dopamine? Is it due to the biological effects of codeine? The author needs to describe the corresponding reasons for selection in the discussion.

Response: Thanks. It is a known fact that testosterone as well as dopamine is essential in sexual competence, and testosterone is one of the regulators of dopamine. This study evaluated dopamine as a neuroendocrine hormone that could alter sexual function outside the hypothalamic-pituitary-testicular axis (Page 5, line 1; page 21). 

6、There are many irregularities in writing in the manuscript:

（1） The notation is not standardized, and no specific explanation is given for each small figure

Response: Thanks. Appropriate legends explaining the notations have been provided per figure and table.

（2） For the reference section, a unified format is required

Response: Thanks. This has been modified for conformity.

.

7、There are many grammatical errors in the full text, which make it difficult to read the manuscript. Please check it.

Response: Thanks. The manuscript has been proof-read by a native speaker and modified.

---

## [Decision Letter · Decision Letter 1]

30 Aug 2022

L-Arginine Reverses Maternal and Pre-pubertal Codeine Exposure-Induced Sexual Dysfunction via Upregulation of Androgen Receptor Gene and NO/cGMP Signaling

PONE-D-22-06170R1

Dear Dr. Ajayi,

We’re pleased to inform you that your manuscript has been judged scientifically suitable for publication and will be formally accepted for publication once it meets all outstanding technical requirements.

Kind regards,

Christopher Torrens

Academic Editor

PLOS ONE

Additional Editor Comments (optional):

Reviewers' comments:

Reviewer's Responses to Questions

**Comments to the Author**

1. If the authors have adequately addressed your comments raised in a previous round of review and you feel that this manuscript is now acceptable for publication, you may indicate that here to bypass the “Comments to the Author” section, enter your conflict of interest statement in the “Confidential to Editor” section, and submit your "Accept" recommendation.

Reviewer #1: (No Response)

Reviewer #2: (No Response)

2. Is the manuscript technically sound, and do the data support the conclusions?

Reviewer #1: Yes

Reviewer #2: Partly

3. Has the statistical analysis been performed appropriately and rigorously? 

Reviewer #1: Yes

Reviewer #2: No

4. Have the authors made all data underlying the findings in their manuscript fully available?

Reviewer #1: Yes

Reviewer #2: Yes

5. Is the manuscript presented in an intelligible fashion and written in standard English?

Reviewer #1: Yes

Reviewer #2: Yes

6. Review Comments to the Author

Reviewer #1: The authors responded to the comments and justified many points, however, I still wonder why the testis histology was not performed (it would not be impacted by the sexual behavior, only the epididymis, due to the sperm transit time). In addition, I also suggest changing how statistical differences are expressed. In the tables and figures, the letters are showing the statistical difference comparing to a specific group (A for control group, B for X group...) leaving everything very polluted and confusing. I strongly advise authors to use letters to express statistical differences between groups (A, B if different from A, AB if similar to A and B...). It's cleaner and easier to understand. On the other hand, in the table where there was no statistical difference, there is no need to put the specifications below (table 03)

Reviewer #2: L-Arginine Reverses Maternal and Pre-pubertal Codeine Exposure-Induced Sexual Dysfunction via Upregulation of Androgen Receptor Gene and NO/cGMP Signaling, the article is special work, can be accepted

7. PLOS authors have the option to publish the peer review history of their article (what does this mean?). If published, this will include your full peer review and any attached files.

Reviewer #1: No

Reviewer #2: **Yes: **no

---

## [Editor Report · Acceptance letter]

4 Sep 2022

PONE-D-22-06170R1 

L-Arginine Reverses Maternal and Pre-pubertal Codeine Exposure-Induced Sexual Dysfunction via Upregulation of Androgen Receptor Gene and NO/cGMP Signaling 

Dear Dr. Ajayi:

I'm pleased to inform you that your manuscript has been deemed suitable for publication in PLOS ONE. Congratulations! Your manuscript is now with our production department. 

Kind regards, 

on behalf of

Dr. Christopher Torrens 

Academic Editor

PLOS ONE